# AutoSurvey: Large Language Models Can Automatically Write Surveys

**Yidong Wang**[1,2*], **Qi Guo**[2,3*],
**Wenjin Yao**[2], **Hongbo Zhang**[1], **Xin Zhang**[4], **Zhen Wu**[3],**Meishan Zhang**[4],
**Xinyu Dai**[3], **Min Zhang**[4], **Qingsong Wen**[5], **Wei Ye**[2†], **Shikun Zhang**[2†], **Yue Zhang**[1†]

[1]Westlake University, [2]Peking University,
[3]Nanjing University, [4]Harbin Institute of Technology, Shenzhen, [5]Squirrel AI

## Abstract

This paper introduces AutoSurvey, a speedy and well-organized methodology for automating the creation of comprehensive literature surveys in rapidly evolving fields like artificial intelligence. Traditional survey paper creation faces challenges due to the vast volume and complexity of information, prompting the need for efficient survey methods. While large language models (LLMs) offer promise in automating this process, challenges such as context window limitations, parametric knowledge constraints, and the lack of evaluation benchmarks remain. AutoSurvey addresses these challenges through a systematic approach that involves initial retrieval and outline generation, subsection drafting by specialized LLMs, integration and refinement, and rigorous evaluation and iteration. Our contributions include a comprehensive solution to the survey problem, a reliable evaluation method, and experimental validation demonstrating AutoSurvey's effectiveness.

## 1 Introduction

Survey papers provide essential academic resources, offering comprehensive overviews of recent research developments, highlighting ongoing trends, and identifying future directions [1, 2, 3, 4]. However, crafting these surveys is increasingly challenging, especially in the fast-paced domain of Artificial Intelligence including large language models(LLMs) [5, 6, 7, 8]. Figure 1a illustrates a significant trend: in just the first four months of 2024 alone, over 4,000 papers containing the phrase "Large Language Model" in their titles or abstracts were submitted to arXiv. This surge highlights a critical academic issue: the rapid accumulation of new information often outpaces the capacity for comprehensive scholarly review and synthesis, emphasizing the growing need for more efficient methods to synthesize the expanding literature. Moreover, as depicted in Figure 1b, while the number of survey papers has rapidly increased, the growing difficulty of producing traditional human-authored survey papers—due to the sheer volume and complexity of data—remains a significant challenge. This challenge is evidenced by the lack of comprehensive surveys in many fields (Figure 1c), which hinders knowledge transfer and makes it difficult for new researchers to efficiently navigate the vast amount of available information.

The advent of LLMs [7, 9] presents a promising avenue for addressing these challenges. These models, trained on extensive text corpora, demonstrate remarkable capabilities in understanding and generating human-like text, even in long-context scenarios [10, 11, 12]. Despite these advancements, the practical application of LLMs to survey generation is fraught with challenges. Firstly, **context**

---

*Equal contribution. yidongwang37@gmail.com, qguo@smail.nju.edu.cn; Yidong Wang did this work during his internship at Squirrel AI.

†Correspondence to: wye@pku.edu.cn, zhangsk@pku.edu.cn, zhangyue@westlake.edu.cn.

38th Conference on Neural Information Processing Systems (NeurIPS 2024).

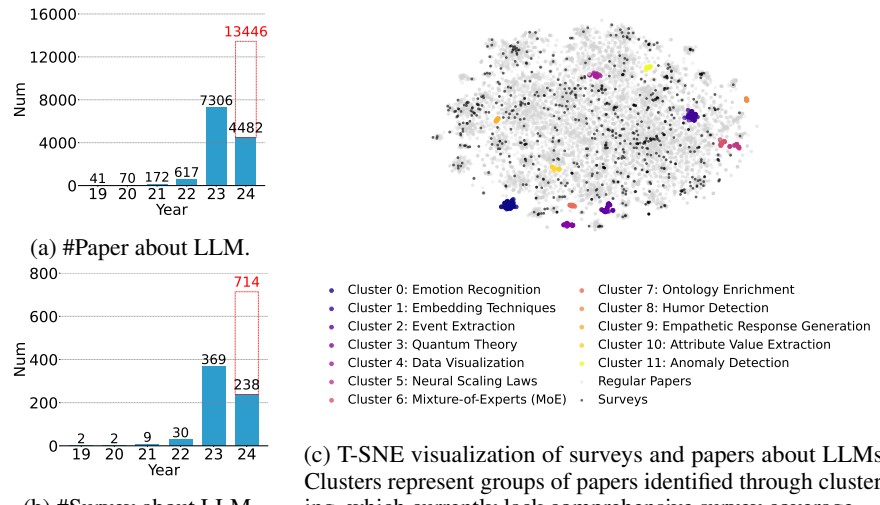

(a) #Paper about LLM.

(b) #Survey about LLM.

(c) T-SNE visualization of surveys and papers about LLMs. Clusters represent groups of papers identified through clustering, which currently lack comprehensive survey coverage.

Figure 1: Depicting growth trends from 2019 to 2024 in the number of LLMs-related papers (a) and surveys (b) on arXiv, accompanied by a T-SNE visualization. The data for 2024 is up to April, with a red bar representing the forecasted numbers for the entire year. While the number of surveys is increasing rapidly, the visualization reveals areas where comprehensive surveys are still lacking, despite the overall growth in survey numbers. The research topics of the clusters in the T-SNE plot are generated using GPT-4 to describe their primary focus areas. **These clusters of research voids can be addressed using AutoSurvey at a cost of $1.2 (cost analysis in Appendix D) and 3 minutes per survey. An example survey focused on Emotion Recognition using LLMs is in Appendix F.**

window limitations: LLMs encounter inherent restrictions in output length due to limited processing windows [13, 14, 15, 16, 17]. While several advanced large models, including GPT-4 and Claude 3, support inputs exceeding 100k tokens, their output is still limited to fewer than 8k tokens (the output length of GPT-4 is 8k, and the output length of Claude 3 is 4k). Writing a comprehensive survey typically requires reading hundreds of papers, resulting in input sizes far beyond the capacity of even the most advanced models. Moreover, a well-written survey itself spans tens of thousands of tokens, making it highly challenging to generate such extensive content directly with large models. Secondly, **parametric knowledge constraints**: Sole reliance on an LLM's internal knowledge is insufficient for producing surveys that require comprehensive and accurate references [18, 19, 20]. LLMs may generate content based on inaccuracies or even non-existent "hallucinated" references. Moreover, these models cannot incorporate the latest studies not included in their training data, which limits the breadth and depth of the surveys they generate. Thirdly, **the lack of evaluation benchmark:** after production, reliable metrics to evaluate the quality of outputs from LLMs are lacking. Relying on human review for quality assessment is not only resource-intensive but also lacks scalability [21, 22, 23]. This presents a significant obstacle to the widespread adoption of LLMs for academic synthesis, where rigorous standards of accuracy and reliability are paramount.

In response to these challenges, we introduce AutoSurvey: a speedy and well-organized methodology for conducting comprehensive literature surveys. Specifically, AutoSurvey's primary innovations include: **logical parallel generation**: AutoSurvey employs a two-stage generation approach to parallelly generate survey content efficiently. Initially, multiple LLMs work concurrently to create detailed outlines. A final, comprehensive outline is then synthesized from these individual outlines, setting a clear framework for content development. Subsequently, each subsection of the survey is generated in parallel and guided by the outline, which significantly accelerates the process. To overcome potential transition and consistency issues due to segmented generation phases, AutoSurvey integrates a systematic revision phase. After the initial parallel generation, each section undergoes thorough revision and polishing, ensuring smooth transitions and enhanced overall document consistency. The sections are then seamlessly merged to produce a cohesive and well-organized final document. **Real-time knowledge update**: AutoSurvey incorporates a Real-time Knowledge Update mechanism using a Retrieval-Augmented Generation (RAG) approach [24, 25, 26]. This feature ensures that every aspect of the survey reflects the most current studies. When a survey topic is input by the user,

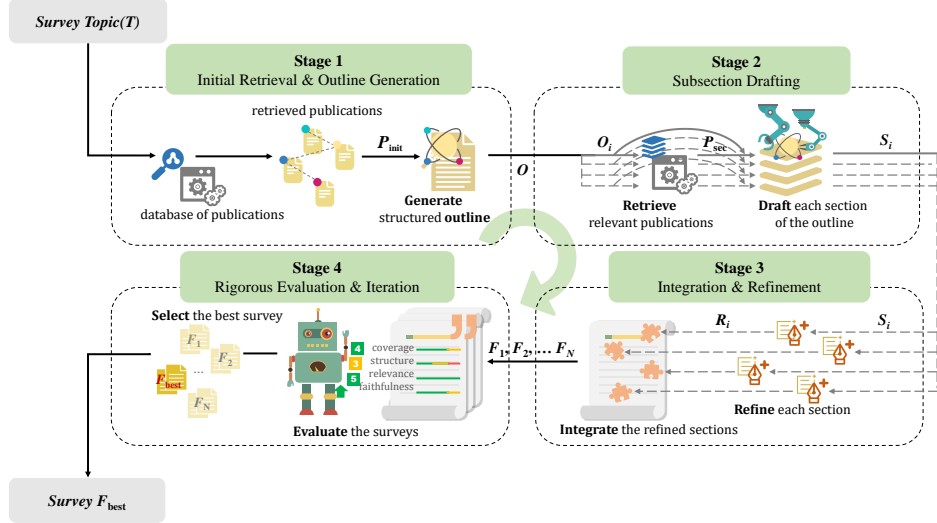

Figure 2: The AutoSurvey Pipeline for Generating Comprehensive Surveys.

AutoSurvey leverages the RAG system to retrieve the latest relevant papers, forming the basis for generating a structured and informed outline. During subsection writing, the system dynamically pulls in new research articles relevant to the specific content under development. This approach ensures that citations are current and the survey content is aligned with the latest developments in the field, significantly enhancing the accuracy and depth of the literature review. **Multi-LLM-as-judge evaluation**: AutoSurvey employs the Multi-LLM-as-Judge strategy, leveraging the LLM-as-Judge method for text evaluation [22, 21, 23]. This approach generates initial evaluation metrics using multiple large language models, which process a substantial corpus of high-quality surveys. These metrics are refined by human experts to ensure precision and adherence to academic standards. The Multi-LLM-as-Judge method assesses generated content across two main dimensions: (1) Citation Quality, verifying the accuracy and reliability of the information presented, with sub-indicators for Recall and Precision. (2) Content Quality, consisting of Coverage (assessing the extent of topic encapsulation), Structure (evaluating logical organization and coherence), and Relevance (ensuring alignment with the main topic). By utilizing multiple LLMs, this strategy minimizes bias and ensures a balanced and comprehensive assessment, upholding rigorous academic standards.

Extensive experimental results across different survey lengths (8k, 16k, 32k, and 64k tokens) demonstrate that AutoSurvey consistently achieves high citation and content quality scores. At 64k tokens, AutoSurvey achieves 82.25% recall and 77.41% precision in citation quality, outperforming naive RAG-based LLMs (68.79% recall and 61.97% precision) and approaching human performance (86.33% recall and 77.78% precision). In content quality at 64k tokens, AutoSurvey scores 4.73 in coverage, 4.33 in structure, and 4.86 in relevance, closely aligning with human performance (5.00, 4.66, and 5.00 respectively). At shorter lengths (8k, 16k, and 32k tokens), AutoSurvey also maintains strong performance across all metrics. Furthermore, the Spearman's rho values indicate a moderate positive correlation between the rankings provided by the LLMs and those given by human experts. The mixture of models achieves the highest correlation at 0.5429, indicating a strong alignment with human preferences. These results reinforce the effectiveness of our multi-LLM scoring mechanism, providing a reliable proxy for human judgment across varying survey lengths.

In conclusion, to the best of our knowledge, AutoSurvey is the first system to explore the potential of large model agents in writing extensive academic surveys. It proposes evaluation criteria for surveys that align with human preferences, providing a valuable reference for future related research.

## 2 Methodology

In this section, we describe the methodology employed by AutoSurvey to automate the creation of comprehensive literature surveys. Our approach systematically progresses through four distinct phases—Initial Retrieval and Outline Generation, Subsection Drafting, Integration and Refinement,

and Rigorous Evaluation and Iteration. Each phase is meticulously designed to address specific challenges associated with survey creation, thereby enhancing the efficiency and quality of the resulting survey document. The pseudo code of AutoSurvey can be found at Algorithm 1.

---

**Algorithm 1** AUTOSURVEY: Automated Survey Creation Using LLMs.

1: **Input:** Survey topic $T$, publications database $D$
2: **Output:** Final refined and evaluated survey document $F_{best}$
3: **for** each survey generation trial $t = 1$ to $N$ **do**
4:    **Phase 1: Initial Retrieval and Outline Generation**
5:    Retrieve initial pool of publications $P_{\text{init}} \leftarrow \text{Retrieve}(T, D)$
6:    Generate outline $O \leftarrow \text{Outline}(T, P_{\text{init}})$
7:    **Phase 2: Subsection Drafting**
8:    **for** each section $O_i$ in $O$ **in parallel do**
9:        Retrieve relevant publications $P_{\text{sec}} \leftarrow \text{Retrieve}(O_i, D)$
10:       Draft subsection $S_i \leftarrow \text{Draft}(O_i, P_{\text{sec}})$
11:   **end for**
12:   **Phase 3: Integration and Refinement**
13:   Refine the merged document to improve coherence $R_i \leftarrow \text{Refine}(S_i)$
14:   Merge subsection drafts into a single document $F_t \leftarrow \text{Merge}(R_1, R_2, \ldots, R_n)$
15: **end for**
16: **Phase 4: Rigorous Evaluation and Iteration**
17: Evaluate and select the best survey document $F_{\text{best}} \leftarrow \text{Evaluate}(F_1, F_2, \ldots, F_N)$
18: **Return:** Refined and evaluated survey $F_{\text{best}}$

---

**Initial Retrieval and Outline Generation**    The process begins with the Initial Retrieval and Outline Generation phase. Utilizing an embedding-based retrieval technique, AutoSurvey scans a database of publications to identify papers most pertinent to the specified survey topic $T$. This phase is crucial for ensuring that the survey is grounded in the most relevant and recent research. The retrieved publications $P_{\text{init}}$ are then used to generate a structured outline $O$, which ensures comprehensive coverage of the topic and logical structuring of the survey. To provide more detailed guidance for writing subsections, the outline generation includes not only titles for each subsection but also brief descriptions. These descriptions convey the main idea of each subsection, aiding in the overall clarity and direction of the survey. Given the extensive amount of relevant papers extracted during this stage, the total length of $P_{\text{init}}$ often exceeds the context window size of the LLM. To address this, papers are randomly divided according to the LLM's context window size, resulting in the creation of multiple outlines. The model then consolidates these outlines to form the final comprehensive outline. Finally, the outline $O$ of the entire survey is represented as $O = \text{Outline}(T, P_{\text{init}})$.

**Subsection Drafting**    With the structured outline in place, the Subsection Drafting phase commences. During this phase, specialized LLMs draft each section of the outline in parallel. This method not only accelerates the drafting process but also ensures detailed and focused content generation for each survey section, adhering to the thematic boundaries established by the outline. When writing the content of each subsection, the sub-outline $O_i$ of that subsection will be used to retrieve the necessary relevant reference papers $P_{\text{sec}}$ to provide information that aligns more closely with the main idea of the subsection. During the writing process, the model is required to cite the provided reference papers to support the generated content. The references in the generated content will be extracted and mapped to the corresponding arXiv papers (see Appendix B for details). The $i_{th}$ subsection $S_i$ can be expressed as: $S_i = \text{Draft}(O_i, P_{\text{sec}})$.

**Integration and Refinement**    Following the drafting phase, each section $S_i$ is individually refined to enhance readability, eliminate redundancies, and ensure a seamless narrative. The refined sections $R_i$ are then merged into a cohesive document $F$, which is essential for maintaining a logical flow and coherence throughout the survey. During the refinement process, the model needs to polish each subsection based on the local context (considering the previous and following subsections) to improve readability, eliminate redundancies, and enhance coherency. Additionally, the model is required to check the correctness of the cited references in the content and correct any errors in the citations. This procedure can be represented by: $F = \text{Merge}(R_1, R_2, \ldots, R_n)$, where $R_i = \text{Refine}(S_i)$.

**Rigorous Evaluation and Iteration** The final phase involves a rigorous evaluation and iteration process, where the survey document is assessed through a Multi-LLM-as-Judge strategy. This evaluation critically examines the survey in several aspects. The insights gained from this evaluation are used to guide further refinements, ensuring the survey meets the highest academic standards. The best survey is chosen from $N$ candidates. The final output of AutoSurvey is $F_{\text{best}} = \text{Evaluate}(\{F_1, F_2, \ldots, F_N\})$.

The methodology outlined here—from initial data retrieval to sophisticated multi-faceted evaluation—ensures that AutoSurvey effectively addresses the complexities of survey creation in evolving research fields using advanced LLM technologies.

## 3 Experiments

**Setup** We conduct comprehensive experiments to evaluate the performance of AutoSurvey, comparing it against traditional methods for generating survey papers. For the drafting phase of AutoSurvey, we utilize Claude-3-Haiku, known for its speed and cost-effectiveness, capable of handling 200K tokens. For evaluations, we employ a combination of GPT-4, Claude-3-Haiku, and Gemini-1.5-Pro[3]. The evaluation covers the following key performance metrics:

- **Survey Creation Speed**: To estimate the time it takes for humans to write a document, we use a mathematical model with the following parameters: $L$ (the length of the document), $E$ (the number of experts), $M$ (the writing speed of each expert), $T_r$ (the preparation time for research and data collection), $T_w$ (the actual writing time, $T_w = \frac{L}{E \times M}$), and $T_e$ (the editing and revision time, $T_e = \frac{1}{2}T_w$). Assuming an ideal situation where $E = 10$, $M = 2000$ tokens/hour, $T_r = 5$ hours, and $T_e = \frac{1}{2}T_w$, the total time $Time$ is calculated as:

$$Time = T_r + T_w + T_e = T_r + \frac{L}{E \times M} + \frac{1}{2} \times \frac{L}{E \times M}. \tag{1}$$

  For Naive RAG and AutoSurvey, we count all the time of API calls. The speed is calculated as $Speed = \frac{1}{Time(\text{hours})}$.

- **Citation Quality**: Adopted from [27], this metric assesses the accuracy and relevance of citations in the survey. Assuming a set of claims $C = \{c_1, c_2, \ldots\}$ extracted from the survey, the metric utilizes an NLI model $h$ to decide whether a claim $c_i$ is supported by its references $\text{Ref}_i = \{r_{i_1}, r_{i_2}, \ldots\}$, where each $r_{i_k}$ represents one paper cited. $h(c_i, \text{Ref}_i) = 1$ means that the references can support the claim, and $h(c_i, \text{Ref}_i) = 0$ otherwise. Refer to Appendix C for more details. Citation quality encompasses two sub-metrics:

  - **Citation Recall**: Measures whether all statements in the generated text are fully supported by the cited passages, which is calculated as

$$\text{Recall} = \frac{\sum_{i=1}^{|C|} h(c_i, \text{Ref}_i)}{|C|}. \tag{2}$$

  - **Citation Precision**: Identifies irrelevant citations, ensuring that the provided citations are pertinent and directly support the statements. Before listing the formula for precision, a function $g$ is defined as $g(c_i, r_{i_k}) = (h(c_i, \{r_{i_k}\}) = 1) \cup (h(c_i, \text{Ref}_i \setminus \{r_{i_k}\}) = 0)$, which measures whether the paper $r_{i_k}$ is related to the claim $c_i$. The precision is

$$\text{Precision} = \frac{\sum_{i=1}^{|C|} \sum_{k=1}^{|\text{Ref}_i|} h(c_i, \text{Ref}_i) \cap g(c_i, r_{i_k})}{\sum_{i=1}^{|C|} |\text{Ref}_i|}. \tag{3}$$

- **Content Quality**: An overarching metric evaluating the excellence of the written survey, encompassing three sub-indicators. Each sub-indicator is judged by LLMs according to a 5-point rubric, calibrated by human experts to meet academic standards. Note that the detailed scoring criteria are provided in Table 1.

  - **Coverage**: Assesses the extent to which the survey encapsulates all aspects of the topic.
  - **Structure**: Evaluates the logical organization and coherence of each section.
  - **Relevance**: Measures how well the content aligns with the research topic.

---

[3]Specifically, we use gpt-4-0125-preview, claude-3-haiku-20240307 and Gemini-1.5-pro-preview.

Table 1: Content Quality Criteria.

| Criteria | Scores |
|---|---|
| **Coverage** | *Score 1*: The survey has very limited coverage, only touching on a small portion of the topic and lacking discussion on key areas.
*Score 2*: The survey covers some parts of the topic but has noticeable omissions, with significant areas either underrepresented or missing.
*Score 3*: The survey is generally comprehensive in coverage but still misses a few key points that are not fully discussed.
*Score 4*: The survey covers most key areas of the topic comprehensively, with only very minor topics left out.
*Score 5*: The survey comprehensively covers all key and peripheral topics, providing detailed discussions and extensive information. |
| **Structure** | *Score 1*: The survey lacks logic, with no clear connections between sections, making it difficult to understand the overall framework.
*Score 2*: The survey has weak logical flow with some content arranged in a disordered or unreasonable manner.
*Score 3*: The survey has a generally reasonable logical structure, with most content arranged orderly, though some links and transitions could be improved such as repeated subsections.
*Score 4*: The survey has good logical consistency, with content well arranged and natural transitions, only slightly rigid in a few parts.
*Score 5*: The survey is tightly structured and logically clear, with all sections and content arranged most reasonably, and transitions between adjacent sections smooth without redundancy. |
| **Relevance** | *Score 1*: The content is outdated or unrelated to the field it purports to review, offering no alignment with the topic.
*Score 2*: The survey is somewhat on topic but with several digressions; the core subject is evident but not consistently adhered to.
*Score 3*: The survey is generally on topic, despite a few unrelated details.
*Score 4*: The survey is mostly on topic and focused; the narrative has a consistent relevance to the core subject with infrequent digressions.
*Score 5*: The survey is exceptionally focused and entirely on topic; the article is tightly centered on the subject, with every piece of information contributing to a comprehensive understanding of the topic. |

**Baselines** We compare AutoSurvey with surveys authored by human experts (collected from Arxiv) and naive RAG across 20 different computer science topics across 20 different topics in the field of LLMs (see Table 7). For the naive RAG, we begin with a title and a survey length requirement, then iteratively prompt the model to write the content until completion. Note that we also provide the model with the same number of reference papers with AutoSurvey. To make a more comprehensive comparison, we additionally introduced two baselines: RAG+Reflection, which involves reflecting the generated survey content from RAG, and RAG+Query Rewriting, where the LLM reformulates the retrieval query based on the topic.

For AutoSurvey, we utilize a corpus of 530,000 computer science papers from arXiv as the retrieval database. During the initial drafting stage, we retrieve 1200 papers relevant to the given topic and split them into several chunks with a window size of 30,000 tokens. The model generates an outline for each chunk and merges these outlines into a final comprehensive outline, using only the abstracts of the papers at this stage. The outline predetermines the number of sections as 8. For subsection drafting, the models generate specific sections using the outline and 60 papers retrieved based on the subsection descriptions, focusing on the main body of each paper (up to the first 1,500 tokens). During the reflection and polishing stage, the same reference papers are provided to the model to ensure consistency and accuracy. The iteration number N is set to 2. Note that human writing surveys used for evaluation are excluded during the retrieval process. For more details of implementations, see Appendix B, and the prompts are presented in Appendix F.

Table 2: Results of Naive RAG, Human writing and AutoSurvey. Both of AutoSurvey and Naive RAG use Claude-haiku as the writer. **Note that human writing surveys used for evaluation are excluded during the retrieval process.**

| Survey Length (#tokens) | Methods | Speed | Citation quality | | Content Quality | | | |
|---|---|---|---|---|---|---|---|---|
| | | | Recall | Precision | Coverage | Structure | Relevance | Avg. |
| 8k | Human writing | 0.16 | 80.00 | 87.50 | 4.50 | 4.16 | 5.00 | 4.52 |
| | Naive RAG | 79.67 | $78.14_{\pm5.23}$ | $71.92_{\pm6.83}$ | $4.40_{\pm0.48}$ | $3.86_{\pm0.71}$ | $4.86_{\pm0.33}$ | 4.33 |
| | Naive RAG+Reflection | - | $82.25_{\pm2.89}$ | $76.84_{\pm2.59}$ | $4.46_{\pm0.37}$ | $4.02_{\pm0.49}$ | $4.86_{\pm0.41}$ | 4.42 |
| | Naive RAG+Query Rewriting | - | $80.99_{\pm3.43}$ | $71.83_{\pm3.59}$ | $4.84_{\pm0.23}$ | $4.05_{\pm0.57}$ | $4.88_{\pm0.19}$ | 4.56 |
| | AutoSurvey | 107.00 | $82.48_{\pm2.77}$ | $77.42_{\pm3.28}$ | $4.60_{\pm0.48}$ | $4.46_{\pm0.49}$ | $4.8_{\pm0.39}$ | 4.61 |
| 16k | Human writing | 0.14 | 88.52 | 79.63 | 4.66 | 4.38 | 5.00 | 4.66 |
| | Naive RAG | 43.41 | $71.48_{\pm12.50}$ | $65.31_{\pm15.36}$ | $4.46_{\pm0.49}$ | $3.66_{\pm0.69}$ | $4.73_{\pm0.44}$ | 4.23 |
| | Naive RAG+Reflection | - | $79.67_{\pm2.94}$ | $73.73_{\pm2.32}$ | $4.57_{\pm0.45}$ | $4.28_{\pm0.59}$ | $4.83_{\pm0.23}$ | 4.55 |
| | Naive RAG+Query Rewriting | - | $77.73_{\pm3.86}$ | $66.29_{\pm4.56}$ | $4.70_{\pm0.31}$ | $3.67_{\pm0.63}$ | $4.79_{\pm0.37}$ | 4.32 |
| | AutoSurvey | 95.51 | $81.34_{\pm3.65}$ | $76.94_{\pm1.93}$ | $4.66_{\pm0.47}$ | $4.33_{\pm0.59}$ | $4.86_{\pm0.33}$ | 4.60 |
| 32k | Human writing | 0.10 | 88.57 | 77.14 | 4.66 | 4.50 | 5.00 | 4.71 |
| | Naive RAG | 22.64 | $79.88_{\pm4.35}$ | $65.03_{\pm8.39}$ | $4.41_{\pm0.64}$ | $3.75_{\pm0.72}$ | $4.66_{\pm0.47}$ | 4.23 |
| | Naive RAG+Reflection | - | $80.50_{\pm3.66}$ | $72.18_{\pm3.31}$ | $4.82_{\pm0.17}$ | $4.08_{\pm0.61}$ | $4.49_{\pm0.44}$ | 4.44 |
| | Naive RAG+Query Rewriting | - | $76.56_{\pm4.38}$ | $65.36_{\pm4.92}$ | $4.61_{\pm0.33}$ | $3.96_{\pm0.65}$ | $4.88_{\pm0.21}$ | 4.45 |
| | AutoSurvey | 91.46 | $83.14_{\pm2.44}$ | $78.04_{\pm3.14}$ | $4.73_{\pm0.44}$ | $4.26_{\pm0.69}$ | $4.8_{\pm0.54}$ | 4.58 |
| 64k | Human writing | 0.07 | 86.33 | 77.78 | 5.00 | 4.66 | 5.00 | 4.88 |
| | Naive RAG | 12.56 | $68.79_{\pm11.00}$ | $61.97_{\pm13.45}$ | $4.4_{\pm0.61}$ | $3.66_{\pm0.47}$ | $4.66_{\pm0.47}$ | 4.19 |
| | Naive RAG+Reflection | - | $73.12_{\pm2.49}$ | $68.36_{\pm3.65}$ | $4.66_{\pm0.31}$ | $4.06_{\pm0.53}$ | $4.76_{\pm0.21}$ | 4.47 |
| | Naive RAG+Query Rewriting | - | $69.77_{\pm5.24}$ | $62.21_{\pm6.73}$ | $4.45_{\pm0.33}$ | $3.88_{\pm0.57}$ | $4.69_{\pm0.29}$ | 4.31 |
| | AutoSurvey | 73.59 | $82.25_{\pm3.64}$ | $77.41_{\pm3.84}$ | $4.73_{\pm0.44}$ | $4.33_{\pm0.47}$ | $4.86_{\pm0.33}$ | 4.62 |

**Main Results**    The results of our experiments comparing human writing, Naive RAG, and AutoSurvey for generating academic surveys are summarized in Table 2. The key findings are:

- *AutoSurvey significantly outperforms both human writing and Naive RAG in terms of speed.* For instance, AutoSurvey achieves a speed of 73.59 surveys per hour for a 64k-token survey, compared to 0.07 for human writing and 12.56 for Naive RAG, highlighting the larger gap in speed for longer context generation.

- *AutoSurvey demonstrates superior citation quality compared to other baselines, with performance close to human writing.* For an 8k-token survey, AutoSurvey achieves citation recall and precision scores of 82.48 and 77.42, respectively, surpassing Naive RAG (78.14 recall, 71.92 precision). While human writing achieves the best performance, ours is close to human's across different lengths. We also observe a significant decline in citation quality for other baselines as the survey length increased, whereas AutoSurvey maintains stable performance. We investigate this phenomenon in our ablation study.

- *AutoSurvey excels in content quality, scoring 4.60 on average for a 16k-token survey.* It achieves 4.66 for coverage, 4.33 for structure, and 4.86 for relevance, matching human writing and surpassing Naive RAG.

The experiments indicate that AutoSurvey provides a balanced trade-off between quality and efficiency. It achieves near-human levels of coverage, relevance, and citation quality while maintaining a significantly lower time cost. While human writing still leads in structure and overall quality, the efficiency and performance of AutoSurvey make it a compelling alternative for generating academic surveys. Naive RAG-based LLM, though effective, falls short in several key areas compared to both human writing and AutoSurvey, making it the least preferred method among the three for generating high-quality academic surveys, particularly in terms of structure, due to the lack of outline.

**Meta Evaluation**    To verify the consistency between our evaluation method and human evaluation, we conduct a meta-evaluation involving human experts and our automated evaluation system. Human experts judge pairs of generated surveys to determine which one is superior. This process, referred to as a "which one is better" game, serves as the golden standard for evaluation. We compare the judgments made by our evaluation method against those made by human experts. Specifically, we provide the experts with the same scoring criteria used in our evaluation for reference. The experts rank the generated 20 surveys, and we compare these rankings with those generated by LLM using Spearman's rank correlation coefficient to measure consistency between human and LLM evaluations.

The results of this meta-evaluation are presented in Figure 3. The table shows the Spearman's rho values, indicating the degree of correlation between the rankings given by each LLM and the human experts. The Spearman's rho values indicate a moderate positive correlation between the rankings provided by the LLMs and those given by the human experts, with the mixture of models achieving the highest correlation at 0.5429. These results suggest that our evaluation method aligns well with human preferences, providing a reliable proxy for human judgment.

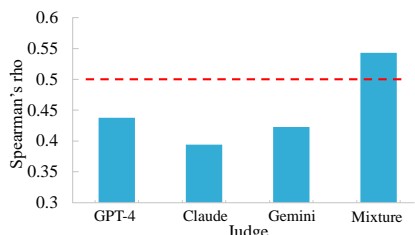

Figure 3: Spearman's rho values indicating the degree of correlation between rankings given by LLMs and human experts. *Note that A value over 0.3 indicates a positive correlation and over 0.5 indicates a strong positive correlation.*

**Ablation study**    To assess the impact of various components on the performance of AutoSurvey, we conduct an ablation study by systematically removing key components: the retrieval mechanism, the reflection phase, and iterations. Additionally, we evaluate the influence of using different base LLMs to demonstrate that even with a less optimal LLM like Claude-haiku, AutoSurvey's performance remains comparable to human-generated surveys.

Table 3 demonstrates that removing the retrieval mechanism significantly degrades citation quality, highlighting its critical role in ensuring accurate and relevant references. The absence of the reflection phase slightly impacts the overall content quality, particularly in structure and coherence.

Table 3: Ablation study results for AutoSurvey with different components removed.

| Methods | Citation Quality | | Content Quality | | | |
| --- | --- | --- | --- | --- | --- | --- |
| | Recall | Precision | Coverage | Structure | Relevance | Avg. |
| AutoSurvey | $83.48_{\pm5.05}$ | $77.15_{\pm6.05}$ | $4.7_{\pm0.45}$ | $4.16_{\pm0.73}$ | $4.93_{\pm0.30}$ | 4.57 |
| AutoSurvey w/o retrieve | $60.11_{\pm6.42}$ | $51.65_{\pm6.33}$ | $4.51_{\pm0.49}$ | $4.01_{\pm0.74}$ | $4.88_{\pm0.32}$ | 4.44 |
| AutoSurvey w/o reflection | $83.23_{\pm3.82}$ | $76.36_{\pm4.08}$ | $4.76_{\pm0.42}$ | $4.13_{\pm0.76}$ | $4.88_{\pm0.32}$ | 4.56 |

Table 4: Performance of AutoSurvey with different base LLM writers.

| Base LLM writer | Citation Quality | | Content Quality | | | |
| --- | --- | --- | --- | --- | --- | --- |
| | Recall | Precision | Coverage | Structure | Relevance | Avg. |
| GPT-4 | $80.25_{\pm4.19}$ | $78.83_{\pm7.00}$ | $4.8_{\pm0.54}$ | $4.46_{\pm0.49}$ | $4.86_{\pm0.33}$ | 4.70 |
| Claude-haiku | $82.45_{\pm2.77}$ | $76.31_{\pm2.18}$ | $4.66_{\pm0.47}$ | $4.26_{\pm0.67}$ | $4.86_{\pm0.33}$ | 4.58 |
| Gemini-1.5-pro | $78.13_{\pm2.39}$ | $71.24_{\pm3.28}$ | $4.86_{\pm0.33}$ | $4.33_{\pm0.78}$ | $4.93_{\pm0.25}$ | 4.69 |
| Human | 85.86 | 80.51 | 4.71 | 4.43 | 5 | 4.70 |

Table 6 shows the performance of AutoSurvey when using different LLMs as the base writer. The results indicate that all three LLMs (GPT-4, Claude-haiku, and Gemini-1.5-Pro) perform well, with GPT-4 slightly outperforming the others in terms of overall content quality. Importantly, even with the less optimal Claude-haiku, AutoSurvey's performance remains competitive with human standards.

Figure 4 presents the effect of different iteration counts on the performance of AutoSurvey. The results show that increasing the number of iterations from 1 to 5 leads to a slight improvement in overall content quality, with diminishing returns after the second iteration.

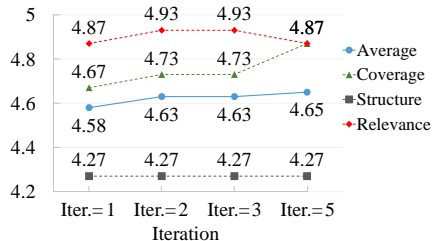

Figure 4: Impact of Iteration on AutoSurvey Performance.

To assess whether the generated survey can provide useful information to enrich the knowledge, we created 50 multiple-choice questions about 5 topics. These questions primarily involve knowledge related to literature, such as identifying which paper proposed a particular method. We compared the accuracy of the Claude model under the following conditions: (1) directly chooses the answer without providing any reference materials, (2) has access to a 32k survey generated by naive RAG-based LLMs, (3) has access to a 32k survey generated by AutoSurvey, and (4) can refer to 20 papers (30k tokens in total) retrieved using the options provided (Upper-bound, directly retrieving the answers).

The results are shown in Table 5 and we find providing topic-related materials can effectively improve the accuracy of answers. Providing option-related papers can be considered an upper bound for performance (73.60%). AutoSurvey improves accuracy by 9.2% compared to directly answering and is 2.4% higher than using naive RAG-based LLM-generated surveys. This demonstrates that our method can effectively provide topic knowledge.

Table 5: Performances given different references.

| Methods | Accuracy |
| --- | --- |
| Direct | $58.40_{\pm4.96}$ |
| Naive RAG-based LLMs | $65.20_{\pm8.06}$ |
| Upper-bound | $73.60_{\pm3.44}$ |
| AutoSurvey | $67.60_{\pm4.96}$ |

As mentioned in the main results, the naive RAG-based generation method demonstrates a notable decline in citation quality as the survey length increases. In contrast, AutoSurvey maintains stable citation quality across varying survey lengths. Such phenomena may be attributed to the streaming generation process, where each step must reference previous content, leading to the accumulation of errors [28]. To validate this, we segment the extracted claims into 20% intervals and calculate the citation recall for each segment. The results in Table 6 indicate that the recall of Naive RAG gradually decreases as the generated text length increases, while AutoSurvey maintains stable performance.

Table 6: Performance of AutoSurvey with different base LLM writers.

| Method | 20% | 40% | 60% | 80% | 100% |
|---|---|---|---|---|---|
| Naive RAG | 76.79 | 73.17 | 71.52 | 64.08 | 49.85 |
| AutoSurvey | 82.86 | 84.89 | 79.04 | 82.27 | 82.29 |

In summary, the ablation study underscores the critical role of the retrieval mechanism and reflection phase in AutoSurvey. Furthermore, the performance is influenced by using different LLMs as the base writer and varying the iteration count. Nevertheless, AutoSurvey consistently performs well across various configurations, showcasing its robustness and efficiency.

**User study**    To evaluate the real-world performance of AutoSurvey, we launch a free trial and distribute an anonymous survey to 141 users. The survey primarily assesses user perceptions of generation quality and whether the generated surveys contribute to their practical work. Users are asked to rate the relevance, structure, and usefulness to their actual work of the generated survey on a scale from 1 (poor) to 5 (excellent). A total of 93 valid responses are collected. Results reveal that users generally perceive AutoSurvey as relevant to topic, well structured, and useful, with most ratings skewed towards 4 and 5. Compared to other aspects, the proportion of scores below 5 in the "Structure" is higher, indicating that there remains room for improvement in further refining the structure. We are pleased to observe that the majority of users think the generated surveys beneficial to their practical work, indicating the utility and practical value of AutoSurvey.

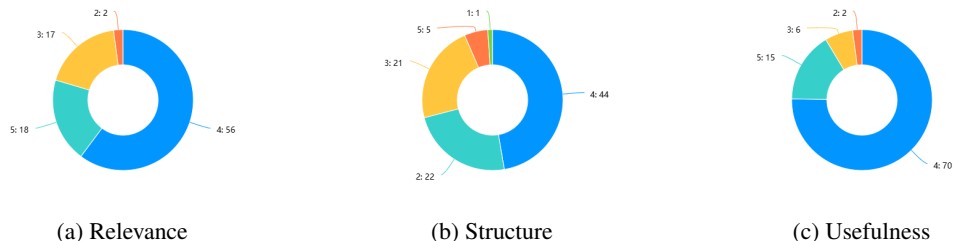

(a) Relevance                    (b) Structure                    (c) Usefulness

Figure 5: Distribution of user ratings in terms of relevance, structure and usefulness.

## 4   Related Work

**Long-form Text Generation**    The ability to effectively process and generate long-form text is a critical challenge for large language models (LLMs) due to the need to maintain coherence and logical flow over extended passages of text [29, 30, 31, 32]. Several works try to address the challenge by directly extending the context window with different Positional Encoding Techniques[33, 34]. However, modifying position encoding strategies requires retraining the model, which is costly. Another solution is using memory-augmented techniques. RecurrentGPT [35] enables the generation of arbitrarily long texts by simulating the recurrence mechanism of RNNs using natural language prompts to store previous contextual information. Temp-Lora [36] enables long text generation by embedding context information into a temporary Lora module updated progressively during generation rather than relying on an extensive context window. These methods effectively establish relationships among tokens and maintain contextual understanding, but still face the issue of long generation times. To further accelerate the generation process, Hierarchical Modeling Techniques have been explored extensively to capture the inherent hierarchical nature of long-form text [37, 38]. Despite such efficiency, it ignores the long dependency of text and may degrade the content quality [39]. To tackle the drawbacks, AutoSurvey, similarly using a Hierarchical generation paradigm, creates a well-organized outline for guidance and refines the generated content to improve the quality.

**Automatic Writing** Due to the high costs associated with manual writing, automated writing has attracted substantial research interest in recent years. Compared to traditional methods, which primarily focus on training models to generate linguistically coherent text [40, 41], the emergency of large language models (LLMs) has opened up new possibilities for automated writing, drawing more attention to broader aspects like faithfulness, logical structure, style, and ethics [42, 43, 44, 45]. For example, Retrieval-Augmented Generation techniques are useful for generating claims with citations [27, 46]. IRP framework [47] generates expository text by iteratively performing content planning, fact retrieval, and paraphrasing to ensure factuality and stylistic consistency. Several works focus on the outline creation to improve the structure of generated content. PaperRobot [48] incrementally writes key elements to generate a paper abstract. STORM [20] designs a refined outline based on multiple rounds of wiki-page-related Q&A to facilitate wiki-like article generations. These methods have only been explored in shorter texts (<4k). In contrast, Autosurvey shows its effectiveness in generating long content (64k), with a focus on academic reviews.

## 5 Limitation

In addition to directly using recall and precision to evaluate citations, we also perform a manual analysis, providing a more comprehensive view of the citation quality. We examine 100 unsupported claims and their corresponding references and find that the errors mainly fall into three categories: (1) Misalignment, (2) Misinterpretation, and (3) Overgeneralization. Misalignment occurs when the connection between them is incorrectly made, such as an irrelevant citation. Misinterpretation happens when the claim and source are related, but the claim incorrectly represents the information from the source. Overgeneralization occurs when a claim extends the conclusions of the source material to a broader context than is supported. Among the three types of errors, overgeneralization accounts for the largest proportion (51%), indicating that LLMs still rely heavily on their parametric knowledge for writing. Misinterpretation has a small proportion (10%), suggesting that LLMs are capable of understanding the content of the references in most cases, avoiding the creation of claims that significantly deviate from the references.

**Misalignment (39%)**: An example is citing the "General Data Protection Regulation (GDPR)" in a context where the referenced paper does not propose GDPR but merely mentions it in the content.

**Misinterpretation (10%)**: An example is claiming that "In-context learning allows LLMs to adapt to new tasks by simply conditioning on a few demonstration examples, without the need for any parameter updates or fine-tuning," based on a paper that focuses on meta-out-of-context learning and mentions the limitations of in-context learning.

**Overgeneralization (51%)**: An example is that "in-context learning can also benefit from advancements in other learning paradigms, such as multi-task learning," based on a paper that discusses multi-task few-shot learning but does not explicitly address its influence on in-context learning.

Among the three types of errors, overgeneralization accounts for the largest proportion (51%), indicating that LLMs still rely heavily on their parametric knowledge for writing. Misinterpretation has a small proportion (10%), suggesting that LLMs are capable of understanding the content of the references in most cases, avoiding the creation of claims that significantly deviate from the references. Additional potential societal impact and ethical considerations are discussed in Appendix E.

## 6 Conclusion

In this paper, we introduce AutoSurvey, a novel methodology leveraging large language models to automate the creation of comprehensive literature surveys. AutoSurvey addresses key challenges such as context window limitations and parametric knowledge constraints through a systematic approach involving initial retrieval, outline generation, parallel subsection drafting, integration, and rigorous evaluation. Our experiments show that AutoSurvey significantly outperforms Naive RAG and matches human performance in content and citation quality, while also being highly efficient. This advancement offers a scalable and effective solution for synthesizing research literature, providing a valuable tool for researchers in rapidly evolving fields like artificial intelligence.

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

# A  Detail of Topics and Human-writing Surveys

We select 20 surveys from different topics within the LLM field. During the selection process, we prioritize both the breadth of the topics and the citation count (from google scholar) of the surveys. The basic information of surveys are listed in Table 7.

Table 7: Survey Table

| Topic | Survey Title | Citations |
|---|---|---|
| In-context Learning | A survey for in-context learning | 323 |
| LLMs for Recommendation | A Survey on Large Language Models for Recommendation | 55 |
| LLM-Generated Texts Detection | A Survey of Detecting LLM-Generated Texts | 42 |
| Explainability for LLMs | Explainability for Large Language Models | 25 |
| Evaluation of LLMs | A Survey on Evaluation of Large Language Models | 183 |
| LLMs-based Agents | A Survey on Large Language Model based Autonomous Agents | 101 |
| LLMs in Medicine | A Survey of Large Language Models in Medicine | 234 |
| Domain Specialization of LLMs | Domain Specialization as the Key to Make Large Language Models Disruptive | 14 |
| Challenges of LLMs in Education | Practical and Ethical Challenges of Large Language Models in Education | 53 |
| Alignment of LLMs | Aligning Large Language Models with Human | 53 |
| ChatGPT | A Survey on ChatGPT and Beyond | 144 |
| Instruction Tuning for LLMs | Instruction Tuning for Large Language Models | 45 |
| LLMs for Information Retrieval | Large Language Models for Information Retrieval | 22 |
| Safety in LLMs | Towards Safer Generative Language Models: Safety Risks, Evaluations, and Improvements | 17 |
| Chain of Thought | A Survey of Chain of Thought Reasoning | 13 |
| Hallucination in LLMs | A Survey on Hallucination in Large Language Models | 116 |
| Bias and Fairness in LLMs | Bias and Fairness in Large Language Models | 12 |
| Large Multi-Modal Language Models | Large-scale Multi-Modal Pre-trained Models | 61 |
| Acceleration for LLMs | A Survey on Model Compression and Acceleration for Pretrained Language Models | 22 |
| LLMs for Software Engineering | Large Language Models for Software Engineering | 49 |

# B  Details of Implementations

We adopt nomic-embed-text-v1.5 [49], a widely used embedding model in RAG applications. To build our database, we store the embeddings of the title and abstract for each paper. Since the context window length is 8k, which is longer than any individual abstract, we embed the raw text directly without chunkings. During generation, related papers are retrieved by the abstract and ranked by their similarity to the query. When generating subsection content, the model needs to write the corresponding paper titles where citations are required. After generation, each title will be embedded as a query and be mapped to the closest paper title in our database. This approach allows the LLMs to use their own parameter knowledge to generate citations without references while ensuring the existence of the generated citations. When calling API, we set temperature = 1 and other parameters as default. Even with the same parameters, the final length of the generated surveys can vary. Therefore, papers with lengths from 8k to 16k are classified into the 8k category, those from 16k to 32k into the 16k category, and so on.

# C  Details of Evaluation

For citation quality, we define a sentence with at least one citation as a claim and extract all the claims from the generated survey. For human evaluations, we invite three PhD students and all of them have experience in writing LLMs-related surveys. We provide them with the same scoring criteria, along with explanations of the specific metrics. They are asked to score based on these criteria, and the final rankings of the generated surveys are determined by the total scores.

## D  Cost Analysis

We present the average number of tokens to generate a 32k-tokens survey, along with the cost of using different LLMs in Table 8.

Table 8: Cost of AutoSurvey

| Input tokens | Output tokens | Claude-haiku | Gemini-1.5-pro | GPT-4 |
|---|---|---|---|---|
| 3009.7K | 112.9K | 0.89$ | 11.72$ | 33.48$ |

## E  Societal Impact and Ethical Considerations

By integrating various specialized databases, our approach can generate academic surveys across different fields, potentially filling the gaps in existing reviews. However, as our method relies on the performance of large models, it inevitably contains citation errors. Therefore, the generated survey content is intended for reference only. All personnel involved in the evaluation process participated voluntarily and received ample compensation. All data used in our experiment is sourced from arXiv and is allowed for non-commercial use.

## F  Prompt used in AutoSurvey

```
ROUGH_OUTLINE_PROMPT =
    '''
    You want to write a overall and comprehensive academic survey
    about [TOPIC].
    You are provided with a list of papers related to the topic below:
    ---
    [PAPER LIST]
    ---
    You need to draft a outline based on the given papers.
    The outline should contains a title and several sections.
    Each section follows with a brief sentence to describe what to
    write in this section.
    The outline is supposed to be comprehensive and contains [SECTION
    NUM] sections.

    Return in the format:
    <format>
    Title: [TITLE OF THE SURVEY]
    Section 1: [NAME OF SECTION 1]
    Description 1: [DESCRIPTION OF SENTCTION 1]

    ...

    Section K: [NAME OF SECTION K]
    Description K: [DESCRIPTION OF SENTCTION K]
    </format>
    The outline:
    '''

SUBSECTION_OUTLINE_PROMPT =
    '''
    You want to write a overall survey about [TOPIC].
    You have created a overall outline below:
    ---
    [OVERALL OUTLINE]
    ---
    The outline contains a title and several sections.
    Each section follows with a brief sentence to describe what to
    write in this section.
```

```
     You need to enrich the section [SECTION NAME].
     The description of [SECTION NAME]: [SECTION DESCRIPTION]
     You need to generate the framwork containing several subsections
    based on the overall outlines.
     Each subsection follows with a brief sentence to describe what to
    write in this subsection.
     These papers provided for references:
     ---
     [PAPER LIST]
     ---
     Return the outline in the format:
     <format>
     Subsection 1: [NAME OF SUBSECTION 1]
     Description 1: [DESCRIPTION OF SUBSENTCTION 1]

     ...

     Subsection K: [NAME OF SUBSECTION K]
     Description K: [DESCRIPTION OF SUBSENTCTION K]
     </format>
     Only return the outline without any other informations:
     '''

MERGING_OUTLINE_PROMPT =
     '''
     You want to write a overall survey about [TOPIC].
     You are provided with a list of outlines as candidates below:
     ---
     [OUTLINE LIST]
     ---
     Each outline contains a title and several sections.
     Each section follows with a brief sentence to describe what to
    write in this section.
     You need to generate a final outline based on these provided
    outlines to make the final outline show comprehensive insights of
    the topic and more logical.
     Return the in the format:
     <format>
     Title: [TITLE OF THE SURVEY]
     Section 1: [NAME OF SECTION 1]
     Description 1: [DESCRIPTION OF SENTCTION 1]

     ...

     Section K: [NAME OF SECTION K]
     Description K: [DESCRIPTION OF SENTCTION K]
     </format>
     Only return the final outline without any other informations:
     '''

SUBSECTION_WRITING_PROMPT =
     '''
     You  wants to write a overall and comprehensive survey about [
    TOPIC].
     You have created a overall outline below:
     ---
     [OVERALL OUTLINE]
     ---
     Below are a list of papers for reference:
     ---
     [PAPER LIST]
     ---

     Now you need to write the content for the subsection:
     "[SUBSECTION NAME]".
```

```
    The details of what to write in this subsection called [SUBSECTION
     NAME] is in this description:
    ---
    [DESCRIPTION]
    ---
    Here is the requirement you must follow:
    1. The subsection is recommended to contain more than [WORD NUM]
    words.
    2. When writing sentences that are based on specific papers above,
     you cite the "paper_title" in a '[]' format to support your
    content.

    Here's a concise guideline for when to cite papers in a survey:
    ---
    1. Summarizing Research: Cite sources when summarizing the
    existing literature.
    2. Using Specific Concepts or Data: Provide citations when
    discussing specific theories, models, or data.
    3. Using Established Methods: Cite the creators of methodologies
    you employ in your survey.
    4. Supporting Arguments: Cite sources that back up your
    conclusions and arguments.
    ---
    Only return the content more than [WORD NUM] words you write for
    the subsection [SUBSECTION NAME] without any other information:
    '''

CITATION_REFLECTION_PROMPT =
    '''
    You want to write a overall and comprehensive survey about [TOPIC
    ].
    Below are a list of papers for references:
    ---
    [PAPER LIST]
    ---
    You have written a subsection below:
    ---
    [SUBSECTION]
    ---
    The sentences that are based on specific papers above are followed
     with the citation of "paper_title" in "[]".
    For example 'the emergence of large language models (LLMs) [PaLM:
    Scaling language modeling with pathways]'

    Here's a concise guideline for when to cite papers in a survey:
    ---
    1. Summarizing Research: Cite sources when summarizing the
    existing literature.
    2. Using Specific Concepts or Data: Provide citations when
    discussing specific theories, models, or data.
    3. Using Established Methods: Cite the creators of methodologies
    you employ in your survey.
    4. Supporting Arguments: Cite sources that back up your
    conclusions and arguments.
    ---

    Now you need to check whether the citations of "paper_title" in
    this subsection is correct.
    Once the citation can not support the sentence you write, correct
    the paper_title in '[]' or just remove it.

    Do not change any other things except the citations.
    Only return the subsection with correct citations:
    '''
```

```
COHERENCY_REFINEMENT_PROMPT =
    ,,,
    You want to write a overall and comprehensive survey about [TOPIC
    ].

    Now you need to help to refine one of the subsection to improve th
     ecoherence of your survey.

    You are provied with the content of the subsection along with the
    previous subsections and following subsections.

    Previous Subsection:
    ---
    [PREVIOUS]
    ---

    Following Subsection:
    ---
    [FOLLOWING]
    ---

    Subsection to Refine:
    ---
    [SUBSECTION]
    ---

    Now refine the subsection to enhance coherence , and ensure that it
     connects more fluidly with the previous and following subsections
    .
    Remember that keep all the essence and core information of the
    subsection intact. Do not modify any citations in [] following the
     sentences!!!!
    Only return the whole refined content of the subsection without
    any other informations:
    ,,,

NLI_PROMPT =
    ,,,
    ---
    Claim:
    [CLAIM]
    ---
    Source:
    [SOURCE]
    ---
    Claim:
    [CLAIM]
    ---
    Is the Claim faithful to the Source?
    A Claim is faithful to the Source if the core part in the Claim
    can be supported by the Source.\n
    Only reply with 'Yes' or 'No':
    ,,,

NAIVE_RAG_PROMPT =
    ,,,
    You are an expert in artificial intelligence who wants to write a
    overall and comprehensive survey about [TOPIC].
    You are provided with a list of papers related to [TOPIC] below:
    ---
    [PAPER LIST]
    ---

    Here is the survey content you have written:
```

```
---
[SURVEY CONTENT]
---

Hers is the requirement of the survey:
1. The survey must be more than [SURVEY LEN] tokens!
2. containing serval sections. Each section contains several
subsections.
5. Cite several paper provided above to support the content you
write.

Here is the format of your writing:
1. '##' indicates the section title
2. '###' indicates the subsection title
3. Only cite the "paper_title" in '[]'. An example of citation: '
the emergence of large language models (LLMs) [Language models are
 few-shot learners; Language models are unsupervised multitask
learners; PaLM: Scaling language modeling with pathways]'

You need to continue writing the survey by adding a new section or
 subsection.

Do not stop until the length of survey is more than [SURVEY LEN]
tokens!!!

Return the content you write:
'''
```

```
# Comprehensive Survey on Emotion Recognition using Large Language
   Models

## 1. Introduction to Emotion Recognition and Large Language Models
   Emotion recognition has been a crucial and active research area in
    the field of affective computing, which aims to enable machines
   to understand, interpret, and respond to human emotions [1].
   Emotions play a fundamental role in human cognition, decision-
   making, and social interaction [2], and the ability to
   automatically recognize and interpret emotions has a wide range of
    applications, including healthcare, education, entertainment, and
    human-computer interaction [3]. The importance of emotion
   recognition is evident in various real-world applications. In
   healthcare, emotion recognition can be used to monitor patient
   mental health, provide personalized therapy, and improve doctor-
   patient communication [4]. In education, emotion recognition can
   help identify student engagement and frustration levels, enabling
   adaptive learning environments that cater to individual needs [5].
    In the entertainment industry, emotion recognition can be used to
    analyze viewer responses and tailor content to evoke desired
   emotional responses [6]. Despite the significant benefits of
   emotion recognition, the field faces several challenges that have
   hindered its widespread adoption and implementation [7]. One of
   the primary challenges is the inherent complexity and subjectivity
    of emotions, which can vary across individuals, cultures, and
   contexts [8]. Emotions are often expressed through multiple
   modalities, including facial expressions, vocal cues, body
   language, and physiological signals, and integrating these diverse
    sources of information is a significant challenge [9].
   Additionally, the availability of high-quality, diverse, and
   annotated emotion datasets is a persistent challenge in the field
   [10]. Many existing datasets are limited in size, lack diversity,
   or have inconsistent or subjective emotion labeling, which can
   lead to biases and poor generalization of emotion recognition
   models [11].
   ...
### 1.1 Background on Emotion Recognition

### 1.2 Large Language Models and their Capabilities

### 1.3 Emotion Representation in LLMs

### 1.4 Multimodal Emotion Recognition using LLMs

## 2. Techniques and Approaches for Emotion Recognition using LLMs

### 2.1 Fine-tuning LLMs on Emotion Datasets

### 2.2 LLM-based Prompting Methods for Emotion Recognition

### 2.3 Integrating LLMs with Other Modalities for Multimodal Emotion
   Recognition

## 3. Enhancing LLM-based Emotion Recognition

### 3.1 Data Augmentation for Improving Emotion Recognition

### 3.2 Prompt Engineering for Emotion Recognition

### 3.3 Integrating External Knowledge for Emotion Recognition

## 4. Challenges, Limitations, and Ethical Considerations

### 4.1 Model Biases and Hallucinations
```

```
### 4.2 Interpretability and Explainability

### 4.3 Ethical Considerations

## 5. Applications and Future Directions

### 5.1 Assistive Robotics

### 5.2 Mental Health Assessment

### 5.3 Customer Service and User Experience

### 5.4 Symbolic Reasoning and Long-tailed Emotions

### 5.5 Robust Evaluation Frameworks

## References

[1] Affective Computing for Large-Scale Heterogeneous Multimedia Data
     A  Survey
[2] Emotion Recognition in Conversation  Research Challenges, Datasets
    , and  Recent Advances
[3] A Comprehensive Survey on Affective Computing; Challenges, Trends,
     Applications, and Future Directions
[4] Affective Computing for Healthcare  Recent Trends, Applications,
    Challenges, and Beyond
[5] Automatic Sensor-free Affect Detection  A Systematic Literature
    Review
[6] Affective Video Content Analysis  Decade Review and New
    Perspectives
[7] Emotion Recognition from Multiple Modalities  Fundamentals and
    Methodologies
[8] The Ambiguous World of Emotion Representation
[9] Multimodal Affective Analysis Using Hierarchical Attention
    Strategy with  Word-Level Alignment
[10] Expression, Affect, Action Unit Recognition  Aff-Wild2, Multi-
    Task  Learning and ArcFace
[11] Feature Dimensionality Reduction for Video Affect Classification
     A  Comparative Study
```

