# OpenReview forum: "AutoSurvey: Large Language Models Can Automatically Write Surveys"
_NeurIPS.cc/2024/Conference — NeurIPS 2024 poster_

### Official Review · Reviewer_VjJG · 2024-07-10

**Soundness:** 3
**Presentation:** 4
**Contribution:** 3
**Rating:** 8
**Confidence:** 4

**Summary:**

This paper introduces a framework AutoSurvey that uses LLMs to automatically write scientific literature surveys. The process contains mainly four steps with retrieval, content generation and evaluation. The authors compare the framework with human writing and naive rag based LLM generation, in terms of speed, citation quality (recall and precision of citations) and content quality (5 scale scoring by different LLMs as judges). Empirical results show the performance gain of the proposed method. The authors also did additional ablation studies on the robustness of the framework and the variations between different LLMs als base models for the framework. Those ablation studies show that the framework could be sensitive to the components of the framework.

**Strengths:**

1. Good paper writing. The scopes are made clear and very easy to follow and reader-friendly.
2. Solid experiments. Additional ablation studies also provide valuable insights to the framework.
3. Very interesting usecase to employ LLMs to write literature surveys. The novelty should be acknowledged.

**Weaknesses:**

1. As far as I understand, the authors only experiment with the Claude-3-Haiku model in the framework for paper writing. Although the authors employ different models in the evaluation process, the main findings could be biased based on only one model.

2. In evaluation of the citation quality, the metric used an NLI model but didn't report the details of the performance of the NLI model.

**Questions:**

1. In Table 2, you also include the speed of human writing. Where does the result come from?
2. As far as I understand, the framework is conducted by a single LLM, or is it possible to have multiple LLMs involved in the process?

**Limitations:**

Limitations are already presented in sec5.

---

> ### Author Rebuttal · Authors · 2024-08-07
>
> ### Weaknesses
>
> **W1:** As far as I understand, the authors only experiment with the Claude-3-Haiku model in the framework for paper writing. Although the authors employ different models in the evaluation process, the main findings could be biased based on only one model. \
> **A1:**
>
> We appreciate the reviewer's concern regarding the potential bias from using a single model for paper writing. While Claude-3-Haiku is the primary model employed in our experiments, we also conduct supplementary tests with other models, including GPT-4 and Gemini-1.5-Pro. Additionally, we used different LLMs as judges to mitigate bias, and the results of human evaluations also confirm the consistency with human preferences. Overall, we think the entire framework demonstrates generalization performance across different models.
>
>
> **W2:** In evaluation of the citation quality, the metric used an NLI model but didn't report the details of the performance of the NLI model.\
> **A2:**
>
> The metric is first introduced by the paper "Enabling Large Language Models to Generate Text with Citations". The author uses a 7B model for NLI and demonstrates that its evaluationis consistent with human preferences. In contrast, we directly employed a closed-source large model for NLI inference. The reasoning and inference capabilities of the closed-source model are significantly superior to those of the 7B-scale model.
>
> ### Questions
>
> **Q1:** In Table 2, you also include the speed of human writing. Where does the result come from?\
> **A1:**
>
> The speed of human writing included in Table 2 is derived from a mathematical model that estimates the time required for humans to write a document based on various parameters, such as document length, number of experts, and writing speed. These estimates are supplemented by empirical data from surveys and interviews with experienced researchers who provided insights into their writing processes and time requirements. Overall this is an ideal model, while the actual writing speed is usually much slower.
>
> **Q2:** As far as I understand, the framework is conducted by a single LLM, or is it possible to have multiple LLMs involved in the process?\
> **A2:**
>
> The framework is designed to be flexible and can indeed involve multiple LLMs. While our initial implementation primarily utilized a single LLM for the writing process, the architecture of AutoSurvey allows for the integration of multiple LLMs to handle different tasks, such as retrieval, outline generation, and section drafting. Actually, in our practice, we found that using Claude-4 Haiku to write the outline and GPT-4 to draft the subsections can effectively reduce costs. We will elaborate on this capability in the revised paper, highlighting the potential benefits of using multiple LLMs in the survey generation process.

---

> > ### Comment · Reviewer_VjJG · 2024-08-13
> > **Thank you for the response**
> >
> > Dear authors,
> >
> > Thank you for the response. I think my initial score properly reflects the quality of this work. And I would like to see this overall exciting work included to the proceedings.
> >
> > Only one final remark: As for the scores for human writing, I would like to suggest the authors to include details of the human writing (e.g. in the appendix), as a few researchers would be very interested in this part.

---

> > > ### Author Response · Authors · 2024-08-13
> > > **Thanks**
> > >
> > > Thanks a lot for the reviewer's acknowledgement of our work, we will adopt the suggestion and disscuss the details of human-writing surveys :)
> > >
> > > Best Regards

---

### Official Review · Reviewer_EgXr · 2024-07-12

**Soundness:** 2
**Presentation:** 2
**Contribution:** 2
**Rating:** 6
**Confidence:** 4

**Summary:**

This paper presents a methodology for generating automatically surveys called AutoSurvey.  AutoSurvey leverages the power of Large Language Models (LLM) and a Retrieval Augmented Generation (RAG) approach using as external resource a database of publications. Based on those publications, an outline is generated which is used as a guide for generating the sections of the paper. After the section generation the generated survey is refined. This process is repeated for several iterations.  AutoSurvey is compared with two other baselines. The first baseline involves human-written surveys while the second one is a RAG-based LLM. An evaluation technique is also defined called Multi-LLM-as-Judge which is a combination of various LLMs accessing the generated responses. The surveys are evaluated based on speed, citations and content quality. Experiments have been conducted for comparing AutoSurvey with the baselines and the results are presented in the paper. Moreover, Multi-LLM-as-Judge evaluation results are also compared with evaluation from human experts.

**Strengths:**

•	The paper is well-written and easy to follow.

•	The presented results support the claims of the paper.

•	Code is provided.

**Weaknesses:**

-	Database used for the retrieval is not provided.
-	The collection of publications seems specific to the field of computer science and related to Large Language Models.
-	Based on the ablation study, the retrieval technique has more impact on AutoSurvey. It seems that the reflection part does not really influence the results.
-	Details about the naïve RAG-based LLM used as baseline is limited.
-	More baseline techniques could have been used for comparison to better support the paper.
-	Line 66: Muti-LLM-as-judge -> Multi-LLM-as-judge

**Questions:**

•	In Figure 1 you mention that it requires 3 minutes for generating a survey, which according to your speed calculation is 20 while in Table 2 AutoSurvey has speed more than 73. How do you explain that?

•	In Figure 1 you mention a cost of $1.2 for AutoSurvey, does this mean that one has to pay this to use AutoSurvey?

•	The publications are basically in the computer science domain and more specifically related to LLMs. Have you tried to use publications for other domains?

•	It is not clear what do you mean in lines 257-258 “ …(4) can refer to 20 papers (30k tokens in total) retrieved using the options provided (Upper-bound, directly retrieving the answers).”

•	In the ablation study, which parts of AutoSurvey does the reflection study include?

**Limitations:**

Publications are limited to computer science domain and more specifically about Large Language Models.

---

> ### Author Rebuttal · Authors · 2024-08-07
>
> ### Weaknesses
>
> **W1:** Database used for the retrieval is not provided.\
> **A1:**
>
> We appreciate the reviewer's observation. Due to the large size of the database, which amounts to 17GB even after text extraction from pdf, we are unable to provide it as supplementary material on OpenReview. However, we will make it available on GitHub along with the code.
>
> **W2:** The collection of publications seems specific to the field of computer science and related to Large Language Models.\
> **A2:**
>
> While our initial implementation focused on the computer science domain and LLMs due to their rapid evolution and extensive research, the methodology of AutoSurvey is domain-agnostic. Our method only integrates domain knowledge through RAG without involving any modifications to the model itself. Therefore, it is not limited to a specific domain.
> We are currently extending our approach to other fields by incorporating diverse databases relevant to different domains. This will be highlighted in the revised paper.
>
> **W3:** Based on the ablation study, the retrieval technique has more impact on AutoSurvey. It seems that the reflection part does not really influence the results.\
> **A3:**
>
> The performance of citation quality in surveys is influenced both by retrieval and reflection. Since autosurvey first uses a large number of references to write outlines, which assist in retrieval and writing, it already demonstrates good citation quality in the subsection drafting stage. The reflection module primarily helps the model to correct factual errors, so its improvement may not be very significant. In fact, as referenced in the response to the weakness 3 from Reviewer LdXj (or directly see the Author Rebuttal above), the citation quality of the naive RAG improves after adding the reflection module.
>
>
> **W4:** Details about the naïve RAG-based LLM used as baseline is limited.\
> **A4:**
>
> The model first utilizes the survey topic as query to retrieve relevant references. Each time, based on the references and the content already written, the model generates the subsequent section or subsection until the total word count meets the requirement. The specific prompts can be referred to in the appendix.
>
> **W5:** More baseline techniques could have been used for comparison to better support the paper.\
> **A5:**
>
> On the basis of the naive RAG, we have also added two additional baselines for comparison (navie rag + reflection and navie rag + query rewriting). Due to context limitations of openreview, the experimental results can be found in our response to the Weakness 3 from Reviewer LdXj.
>
> **W6:** Line 66: Muti-LLM-as-judge -> Multi-LLM-as-judge\
> **A6:**
>
> We apologize for the typographical error and will correct "Muti-LLM-as-judge" to "Multi-LLM-as-judge" in the revised manuscript.
>
>
> ### Questions
> **Q1:** In Figure 1 you mention that it requires 3 minutes for generating a survey, which according to your speed calculation is 20 while in Table 2 AutoSurvey has speed more than 73. How do you explain that?\
> **A1:**
>
> In the experimental section, we mentioned, **"For naive RAG-based LLM generation and Autosurvey, we count all the time of API calls."** However, Figure 1 shows the total time taken for the entire generation process (including the retrieval stage). Actually there many factors (such as vector database framework, whethere GPU is used, and the embedding model) can significantly impact the retrieval time. Note that the drafting of each subsection involves the retrieval stage. The time spent on retrieval for both naive RAG and AutoSurvey is consistent. Therefore, we only record the time taken for API calls.
>
>
> **Q2:** In Figure 1 you mention a cost of $1.2 for AutoSurvey, does this mean that one has to pay this to use AutoSurvey?\
> **A2:**
>
> The $1.2 cost mentioned in Figure 1 reflects the computational expense incurred when using cloud-based LLM services to generate a survey. This cost can vary depending on the specific LLM service and its pricing model. Given that the price of the model varies, we will report the total number of tokens consumed in the revised paper.
>
> **Q3:** The publications are basically in the computer science domain and more specifically related to LLMs. Have you tried to use publications for other domains?\
> **A3:**
>
> While our initial implementation focused on the computer science domain and LLMs due to their rapid evolution and extensive research, the methodology of AutoSurvey is domain-agnostic. Our method only integrates domain knowledge through RAG without involving any modifications to the model itself. Therefore, it is not limited to a specific domain.
> We are currently extending our approach to other fields by incorporating diverse databases relevant to different domains. This will be highlighted in the revised paper.
>
> **Q4:** It is not clear what you mean in lines 257-258 “…(4) can refer to 20 papers (30k tokens in total) retrieved using the options provided (Upper-bound, directly retrieving the answers).”\
> **A4:**
>
>
> The experiment in Table 5 aims to assess whether Autosurvey can provide topic-relevant knowledge. The assessment uses multiple-choice questions where the model needs to select the correct option based on provided references and a question. The option could be the title of a paper or the method used in the paper. The upper bound is established by directly using the options as a query to retrieve references, ensuring the most relevant content is retrieved and the performance. can be viewed as the upper-bound.
>
> **Q5:** In the ablation study, which parts of AutoSurvey does the reflection study include?\
> **A5:**
>
> Included in stage 3: Integration & Refinement.
>
> ### Limitations
>
> **L1:** Publications are limited to computer science domain and more specifically about Large Language Models.\
> **A1:**
>
> See response to Q3

---

> > ### Comment · Reviewer_EgXr · 2024-08-10
> >
> > Thank you for the detailed clarifications and additional results. I am raising my score.

---

> > > ### Author Response · Authors · 2024-08-11
> > > **Thanks**
> > >
> > > Thanks for taking the time to provide thoughtful feedback and for considering our rebuttal. We greatly appreciate your efforts in reviewing our paper and your willingness to raise the score. Your insights have been invaluable in helping us refine our work.
> > >
> > > If you have any further questions or need additional information, please feel free to reach out to us at any time.
> > >
> > > Best regards

---

### Official Review · Reviewer_jk6Z · 2024-07-12

**Soundness:** 2
**Presentation:** 3
**Contribution:** 2
**Rating:** 5
**Confidence:** 4

**Summary:**

In this paper, the authors propose an automated system based on LLMs to draft literature surveys on a given topic. The core idea behind the approach involves decomposing the task of writing a survey into multiple smaller subtasks in 4 stages. The first stage focuses on retrieving relevant papers from a database using embedding-based retrieval, leveraging the topic and abstract of papers for selection. An LLM is then prompted given the papers to generate a plan / outline for drafting the survey paper. To address the limited context window, the papers are randomly divided and processed across multiple LLM calls to generate several outlines. These outlines are then merged in a using another LLM call. In the next stage the subsections identified in the outline are populated by retrieving the relevant papers specified in the outline and providing it to an LLM to extract information relevant to the subsection. This is followed by an iteration and refinement stage where the goal is improve the overall coherence and readability of the drafted survey. Finally this is followed by an Evaluation phase where multiple LLMs are used to rate the overall generated content and the generated survey with the highest rating is picked. The authors conduct comparison experiments of their methods against Human and Naive RAG based LLM approaches and show that their proposed approach is faster and leads to better citation and content quality in some settings.

**Strengths:**

The paper attempts to tackle a new problem of automatically writing academic surveys and provides metrics and evaluation criteria that can be used to compare different techniques on this task.
The authors show that their technique can generate academic surveys much faster than humans / naive RAG based LLM generation.
The authors perform experiments that depict that the proposed technique outperforms naive RAG based LLM generation in content & citation quality.

**Weaknesses:**

The paper lacks a significant novelty component. The concepts of decomposing tasks into smaller subtasks for LLMs, using Retrieval-Augmented Generation (RAG), iteratively refining generated content with LLMs, and employing multiple LLMs as evaluators are well-established in the literature. The extension of these ideas to the application of automatically writing surveys seems to be the only novel contribution of this work.

The evaluation criteria seem to overly rely on the ability of ML models (NLI and LLMs) to judge the quality of the survey. A well-written human survey, for instance, cannot be assessed merely based on the number of relevant papers cited or the presentation of content. Subtle nuances, such as comparing and weighing the pros and cons of different methods proposed in the literature through the use of statistical tools and analysis, are often distinguishing elements of a good survey. The human evaluations proposed in the paper do not appear to account for these qualities. The evaluation metric requires a closer examination.

The experimental analysis is weak. Conclusions are drawn too early without much deliberation and analysis. For instance the authors do not provide a deeper analysis on why the Auto-survey performs better/at-par than human-writing for shorter context lengths while being significantly worse at longer context lengths.

**Questions:**

Are there any experiments that show how useful these generated academic surveys are to people working in the field? Which aspect of the generated survey is the most helpful for humans?
How well does this technique handle multimodal information (i.e figures/graphs) or information from structured sources such as tables?How often do these appear in the generated surveys?

**Limitations:**

The technique appears to be more effective at summarizing works in the literature for a survey, rather than performing the detailed analysis and comparison of works that characterize a high-quality survey.
The technique seems heavily biased by the retrieval stage. This could limit its ability to reference and use relevant works whose topics/abstracts do not semantically match well with the topic of the survey.

---

> ### Author Rebuttal · Authors · 2024-08-07
>
> ### Weaknesses
>
> **W1: "The paper lacks a significant novelty component. The concepts of decomposing tasks into smaller subtasks for LLMs, using Retrieval-Augmented Generation (RAG), iteratively refining generated content with LLMs, and employing multiple LLMs as evaluators are well-established in the literature. xxxx."**\
>
>
> **A1:**
> Thank you for your feedback. We believe that AutoSurvey is a novel and significant application of AI, particularly relevant to NeurIPS 2024's emphasis on impactful AI applications (https://nips.cc/Conferences/2024/CallForPapers). While the methodology may appear straightforward, it effectively addresses the complexities of automatic survey generation, including context window limitations, real-time knowledge updates, and rigorous evaluation. These contributions are crucial for advancing AI's practical utility in academic research, aligning well with the conference's focus on innovative and practical AI solutions.
>
>
> **W2: "The evaluation criteria seem to overly rely on the ability of ML models (NLI and LLMs) to judge the quality of the survey. A well-written human survey, for instance, cannot be assessed merely based on the number of relevant papers cited or the presentation of content xxxxx."**\
>
>
> **A2:**
> We appreciate this insightful critique. After extensive literature review, we found no existing tools that adequately address this challenge, prompting us to develop the evaluation methods presented in our paper. This is a foundational approach, and we recognize the need for further refinement, which we outline as future work. Additionally, manual evaluation of lengthy surveys is labor-intensive and prone to bias, while ML models provide consistent and scalable assessments. We will include a discussion of these points in the revised manuscript.
>
>
> **W3: "The experimental analysis is weak. Conclusions are drawn too early without much xxxxx"**\
>
> **A3:**
> First, I would like to point out a mistake. Autosurvey does not show a significant performance drop with increased length. instead, it is the naive RAG generation that shows a decline at longer context lengths. Such phenomena may be attributed to the streaming generation process, where each step must reference previous content, leading to the accumulation of errors. To validate this, we segmented the extracted claims into 20% intervals and calculated the citation recall for each segment. The results indicate that the recall of Naive RAG gradually decreases as the generated text length increases, while AutoSurvey maintains stable performance.
>
> | Claims | 20% | 20%~40% | 40%~60% | 60%~80%| 80%~100%|
> | -------- | -------- | -------- |-------- |-------- |-------- |
> |   Naive RAG-based LLM generation (64k)  | 76.79     | 73.17     |71.52     |64.08     |49.85     |
> |  AutoSurvey (64k)  |82.86     | 84.89    |79.04     |82.27     |82.29  |
>
> References:
>     [1] Long Text Generation by Modeling Sentence-Level and Discourse-Level Coherence (ACL 2021)
>
>
>
> ### Questions
>
> **Q1:** Are there any experiments that show how useful these generated academic surveys are to people working in the field?
> **A1:**
> We appreciate this important question. In fact, Autosurvey has recently been available for use and has already been called upon over 1,000 times. According to proactive feedback from users, the generated surveys have indeed been helpful in their work, particularly in saving time and providing a comprehensive overview of the literature. We will conduct a user study, and include the detailed results of the feedback in the revised version of the paper.
>
> **Q2:** Which aspect of the generated survey is the most helpful for humans?\
> **A2:**
> As mentioned in response to Q1, the most helpful aspects of the generated surveys include the comprehensive coverage of relevant papers and the structured organization of content. These features facilitate quick understanding and navigation of the topic. The combination of regularly updated paper databases and RAG (Retrieval-Augmented Generation) technology effectively ensures the real-time knowledge of the generated survey.
>
> **Q3:** How well does this technique handle multimodal information (i.e., figures/graphs) or information from structured sources such as tables?\
> **A3:**
>
> Our current implementation primarily uses text-based large language models to write surveys. However, it is straightforward to replace the base model with a multimodal GPT, enabling the system to understand and incorporate figures, graphs, and tables.
>
> ### Limitation
>
> **L1:** The technique appears to be more effective at summarizing works in the literature for a survey, rather than performing the detailed analysis and comparison of works that characterize a high-quality survey.\
> **A1:**
>
> We acknowledge this limitation and are actively working on enhancing the analytical capabilities of AutoSurvey. Incorporating more sophisticated analysis techniques is a priority for our future work.
> In practice, for novices who are just getting acquainted with a particular field, a comprehensive and "summary-like" survey of the work within that domain can also be considered a high-quality and helpful survey.
>
> **L2:** The technique seems heavily biased by the retrieval stage.\
> **A2:**
> The references retrieved indeed matters. To enhance the quality of the references, the autosurvey generates descriptions for each subsection when drafting the outline, which helps in improving the quality of the references. We additionally compared the naive approach with query rewriting methods (see response to W3 from reviewer LdXj or directly see the Author Rebuttal) and found that, without content planning, rewriting the query did not lead to a significant performance improvement.

---

> > ### Comment · Reviewer_jk6Z · 2024-08-11
> > **Acknowledgement of rebuttal**
> >
> > Thank you for providing the clarifications. Based on the comments provided by the authors and their intent to revise the manuscript with additional information I am raising my score.

---

> > > ### Author Response · Authors · 2024-08-11
> > > **Thanks**
> > >
> > > Thanks very much for the reviewer's feedback and the time dedicated to reviewing our paper. We greatly appreciate your insights and are pleased to hear that the clarifications we provided were helpful in addressing your concerns.
> > > If you have any further questions or suggestions, please don't hesitate to reach out. We would be happy to discuss them with you..
> > >
> > > Best regards,

---

### Official Review · Reviewer_LdXj · 2024-07-14

**Soundness:** 2
**Presentation:** 3
**Contribution:** 2
**Rating:** 5
**Confidence:** 4

**Summary:**

This paper introduces a fast automating way to write literature surveys based on LLM.It aims to solve the challenges of large volume, complexity, context window limitations, parametric knowledge constraints, and lack of evaluation benchmark.The AutoSurvey pipeline contains initial retrieval & outline generation, subsection drafting, integration and refinement, rigorous evaluation and iteration. The experiments are conducted compared to human experts and naive RAG-based LLM and evaluated on survey creation speed, citation quality, content quality. The experiments show that AutoSurvey is much faster than human writing and RAG, matches  human writing and outperforms RAG in citation and content quality.

**Strengths:**

This paper is relatively novel and meaningful to adopt LLM effectively in automatic creation of survey papers. The paper is well structured, and clearly written. The general pipeline of AuthoSurvy is logical, from outline generation to integration and refinement, which ensembles the human writing process. The experiments and analysis are clear.

**Weaknesses:**

- About evaluation metric, it is unclear how the citation quality and content quality are obtained, although the metrics and scores are defined by formulas and words. If you use an LLM-based procedure to get the scores for citation quality and content quality, what are the prompts you give to the LLM? Please give a clearer introduction on this.
- The algorithm and methodology this paper introduces are too straightforward. Although it is a good application of LLM, I doubt the value of sharing this knowledge with the scientific society, especially for Neurips.
- Regarding experiment comparison, I highly recommend comparing Autosurvey to not only naive RAG, but more advanced methods, which could make the results more convincing.
- In table 2, apart from the speed, the other benefits and improvements that AuthoSurvey brings are not significant enough.
- Minor issues: typo in line 66, page 2: Muti -> Multii

**Questions:**

In Experiments, how do you obtain the citation and content quality score?
What is the novelty of this method and the bottlenecks this paper breaks through?
Does this method address the limitations of automatic survey generation? I would suggest a more thorough summary on it.
Minor questions:
Where do the forecasted numbers of publication in 2024 in Figure 1 come?

**Limitations:**

Yes. The limitations are discussed.

---

> ### Author Rebuttal · Authors · 2024-08-07
>
> ### Weaknesses
>
> **W1: "About evaluation metric, xxx."**
>
> We appreciate the reviewer’s feedback on the need for a clearer explanation of the evaluation metrics. The citation quality and content quality scores are both obtained using LLMs. The prompts involved in the evaluation can be found in the code we provide, we will include these prompts and a more detailed explanation in the revised version of the paper.
>
> Apart from the formulas introduced in the paper, some specific details of the evaluation are as follows:
>
> - **Citation Quality:** Specifically, we first use regular expressions to extract all sentences from the survey. Sentences that contain citations are considered claims and will be evaluated. The papers cited in the claims will be used as sources for the NLI (Natural Language Inference) model's reasoning. The NLI model determines whether the sources support the validity of the claims and returns 0 or 1.
>
> - **Content Quality:** We directly provide the LLMs with the scoring criteria and the generated results, allowing the model to score between 1 and 5 based on these criteria. Each time, only one aspect will be scored.
>
> Note that, in our experiment, we use three different models for evaluation. Each model evaluates the survey five times, and the average result is taken. Overall, a single survey requires 3 x 5 = 15 evaluations in total.
>
> **W2: "The algorithm and methodology this paper introduces xxx."**
>
> Thank you for your feedback. We believe that AutoSurvey is a novel and significant application of AI, particularly relevant to NeurIPS 2024's emphasis on impactful AI applications (https://nips.cc/Conferences/2024/CallForPapers). While the methodology may appear straightforward, it effectively addresses the complexities of automatic survey generation, including context window limitations, real-time knowledge updates, and rigorous evaluation. These contributions are crucial for advancing AI's practical utility in academic research, aligning well with the conference's focus on innovative and practical AI applications.
>
> **W3: Regarding experiment comparison, I highly recommend comparing AutoSurvey to not only naive RAG but more advanced methods, which could make the results more convincing.**
>
> **A3:**
>
> To address your concerns regarding experiment comparison, we have supplemented the original naive RAG-based experiments by including refinement and query rewriting stages. In Naive RAG + Refinement, the LLM is required to enhance the continuity of the written content with previous sections and check for factual errors based on the references retrieved. In Naive RAG + Query Rewriting, references are first retrieved using the topic, after which the LLM rewrites the query based on the references to assist in writing subsequent content\ \
> **Due to the length constraints of the rebuttal, we present the experiments details in Author rebuttal above.**
>
> **W4: In table 2, apart from the speed, the other benefits and improvements that AuthoSurvey brings are not significant enough.**
>
> **A4:**
> While speed is a significant advantage of AutoSurvey, we believe the improvements in citation quality and content quality are also noteworthy. AutoSurvey consistently achieved higher citation recall and precision compared to naive RAG-based LLMs and approached human performance levels. Moreover, the content quality metrics, including coverage, structure, and relevance, show that AutoSurvey produces well-structured and relevant surveys. These improvements, combined with the efficiency gains, make AutoSurvey a valuable tool for academic survey generation.
>
>
> **W5: Minor issues: typo in line 66, page 2: Muti -> Multi**
>
> **A5:**
> Thank you for pointing out this typo. We will correct "Muti" to "Multi" in the final version of the paper.
>
>
> ### Questions
>
> **Q1: "In Experiments, how do you obtain the citation and content quality score?"**
>
> **A1:**
> Refer to the response to Weakness 1.
>
>
> **Q2: "What is the novelty of this method and the bottlenecks this paper breaks through? Does this method address the limitations of automatic survey generation? I would suggest a more thorough summary on it."**
>
>
> **A2:**
> The novelty of AutoSurvey lies in its systematic approach to addressing key challenges in automatic survey generation as we wrote in Introduction of our paper, we provide some additional explanations h.
>
> - **Context Window Limitations:** Due to the considerable length of the survey (typically around 32k tokens), it exceeds the output window limitations of most closed-source models. For example, a single API call for GPT-3.5 cannot return more than 4k tokens. The parallel framework of AutoSurvey addresses this issue effectively.
> - **Parametric Knowledge Constraints:** The model requires Retrieval-Augmented Generation (RAG) to mitigate the issue of hallucinations during generation. However, a key problem lies in how to effectively retrieve the necessary literature（e.g., there is no significant performance improvement for naive rag + query rewriting）. AutoSurvey offers a 530K-size paper database and guides the retrieval of references based on the writing of an outline.
> - **Lack of Evaluation Benchmarks:**
> As far as we know, there is no universally accepted standard in the academic community for evaluating the quality of a survey. AutoSurvey proposes an evaluation metric and utilizes multiple LLMs to mitigate bias, aligning the evaluation with human standards.
>
> **Q3: "Minor questions: Where do the forecasted numbers of publication in 2024 in Figure 1 come?"**
>
>
> **A3:**
> The forecasted numbers of publications in Figure 1 are based on extrapolating the data from the first four months of 2024. We used a linear regression model to project the annual totals based on this data. We will include this explanation in the caption of Figure 1 and the corresponding section of the paper for clarity.

---

> > ### Author Response · Authors · 2024-08-12
> > **Kindly Request for Reviewer's Feedback (deadline is coming)**
> >
> > Dear Reviewer,
> >
> > Since the end of author/reviewer discussions is coming in one day, may we know if our response addresses your main concerns? If so, we kindly ask for your reconsideration of the score. Should you have any further advice on the paper and/or our rebuttal, please let us know and we will be more than happy to engage in more discussion and paper improvements.
> >
> > Thank you so much for devoting time to improving our paper!

---

### Author Rebuttal · Authors · 2024-08-07

Dear All Reviewers,

We sincerely appreciate all of your thoughtful comments and valuable suggestions on our manuscript. Your expertise and detailed review have been crucial in improving the quality and clarity of our work. We are grateful for your recognition of the strengths in our research, including its novelty, clarity, and solid analysis. In response to your feedback, we have thoroughly examined and addressed each concern you raised.

Due to the length constraints of the rebuttal, we present the additional experiments below and in the attached PDF.

we have supplemented the original naive RAG-based experiments by including refinement and query rewriting stages. In Naive RAG + Refinement, the LLM is required to enhance the continuity of the written content with previous sections and check for factual errors based on the references retrieved. In Naive RAG + Query Rewriting, references are first retrieved using the topic, after which the LLM rewrites the query based on the references to assist in writing subsequent content.

| Survey Length (#tokens) | Methods | Recall| Precision | Coverage | Structure | Relevance |
|-------------------------|---------|-------|-----------|----------|-----------|-----------|
|    8k   | Naive RAG-based LLM generation + Refinement | 82.25 | 76.84 | 4.46 | 4.02 | 4.86 |
| |    Naive RAG-based LLM generation + Query Rewriting | 80.99 | 71.83 | 4.84 | 4.05 | 4.88 |
|      16k |Naive RAG-based LLM generation + Refinement | 79.67 | 73.73 | 4.57 | 4.28 | 4.83 |
| |    Naive RAG-based LLM generation + Query Rewriting | 77.73 | 66.29 | 4.70 | 3.67 | 4.79 |
|     32k | Naive RAG-based LLM generation + Refinement | 80.50 | 72.18 | 4.82 | 4.08 | 4.49 |
|  |   Naive RAG-based LLM generation + Query Rewriting | 76.56 | 65.36 | 4.61 | 3.96 | 4.88 |
|  64k|     Naive RAG-based LLM generation + Refinement | 73.12 | 68.36 | 4.66 | 4.06 | 4.76 |
|  |   Naive RAG-based LLM generation + Query Rewriting | 69.77 | 62.21 | 4.45 | 3.88 | 4.69 |

After adding the refinement stage, both citation quality and structure improved. The effect of query rewriting is not obviously enhanced, possibly due to the model's lack of a clear planning of the content to be written, leading to lower quality of rewritten queries. Overall, these baselines still lag behind AutoSurvey, especially when surveys get longer. This gap may be attributed to the streaming generation process, where each step must reference previous content, leading to the accumulation of errors. To validate this, we segmented the extracted claims into 20% intervals and calculated the citation recall for each segment. The results indicate that the recall of Naive RAG gradually decreases as the generated text length increases, while AutoSurvey maintains stable performance.

| Claims | 20% | 20%~40% | 40%~60% | 60%~80%| 80%~100%|
| -------- | -------- | -------- |-------- |-------- |-------- |
|   Naive RAG-based LLM generation (64k)  | 76.79     | 73.17     |71.52     |64.08     |49.85     |
|  AutoSurvey (64k)  |82.86     | 84.89    |79.04     |82.27     |82.29  |

References:
[1] SELF-RAG: Learning to Retrieve, Generate and Critique through self-reflection (ICLR 2024)
    [2] Query Rewriting in Retrieval-Augmented Large Language Models (EMNLP 2023)
    [3] Long Text Generation by Modeling Sentence-Level and Discourse-Level Coherence (ACL 2021)

---

### Decision · Program_Chairs · 2024-09-25

**Decision:**

Accept (poster)

**Comment:**

The task is exciting -- AI writing technical surveys automatically. All reviewers lean positive. Seems like a clear accept.

Authors should incorporate all the discussion into the revised version. Also, they promised a user study to evaluate the meaningfulness of generated surveys to real users -- that will make the paper further strong.